# Self- versus clinician-collected swabs in anal cancer screening: A clinical trial

Clare E. F. Dyer[1], Fengyi Jin[1], Jennifer M. Roberts[2], I. Mary Poynten[1], Annabelle Farnsworth[2], Leon P. McNally[3], Philip H. Cunningham[4], Andrew E. Grulich[1], Richard J. Hillman[1,5]*

1 Kirby Institute, University of New South Wales, Sydney, NSW, Australia, 2 Douglass Hanly Moir Pathology, Sydney, NSW, Australia, 3 NSW State Reference Laboratory for HIV, St Vincent's Hospital, Sydney, NSW, Australia, 4 St Vincent's Centre for Applied Medical Research, St Vincent's Hospital, Sydney, NSW, Australia, 5 Dysplasia and Anal Cancer Services, St Vincent's Hospital, Sydney, NSW, Australia

* richard.hillman@unsw.edu.au

## Abstract

### Background

Risk of anal cancer is high in certain populations and screening involves collection of anal swabs for HPV DNA and/or cytology testing. However, barriers exist, such as the need for an intimate examination, and stigma around HIV status, sexual orientation, and sexual practices. Self-collected anal swabs (SCA) are a proposed alternative to clinician-collected swabs (CCA) to overcome these barriers.

### Methods

Participants were order-randomised to undergo SCA or CCA first, with a second swab taken immediately afterwards. Sample adequacy was assessed for HPV DNA and cytology testing. CCA was used as the gold standard to calculate sensitivity and specificity of SCA for cytology and HPV results. Acceptability of swab collection was assessed following the procedure.

### Results

There was no significant difference in sample validity for HPV DNA testing between SCA and CCA (p = 0.564). Concordance was >90% for detection of any HR-HPV and HPV16. There was no significant difference in cellular adequacy for cytological testing between SCA and CCA, (p = 0.162). Concordance for cytologic prediction was 88.2% for any cytologic abnormality. Almost half (48.5%) of participants expressed no preference for SCA versus CCA; 15.2% preferred SCA and 35.4% CCA.

### Conclusions

SCA may be an acceptable and feasible alternative to CCA for detecting HPV and cytological abnormalities in a clinic population.

**Data Availability Statement:** All relevant data are within the manuscript and its Supporting Information files.

**Funding:** We would like to thank the St Vincent's Clinic Foundation for funding this project. The funders had no role in study design, data collection and analysis, decision to publish, or preparation of the manuscript.

**Competing interests:** The authors have declared that no competing interests exist.

## Introduction

Anal cancer is a highly stigmatised condition, associated with substantial morbidity and mortality [1, 2]. Anal squamous cell carcinoma (SCC) occurs at low rates in the general population, at 1–2 cases per 100,000 population per year, but incidence rates are much higher in certain sub-populations, such as gay, bisexual and other men who have sex with men (GBM) [3]. GBM living with HIV have the highest incidence rate of 89 cases/100,000 population [4–6].

There is a strong causal relationship between infection with high-risk human papillomavirus (HR-HPV), particularly genotype HPV16, and development of anogenital cancers including of the vagina, vulva, cervix, and anus [7, 8]. Persistent HR-HPV infection can lead to development of high-grade squamous intraepithelial lesions (HSIL)–the precursor to anal SCC. The recent ANal Cancer/HSIL Outcomes Research (ANCHOR) study demonstrated that treating anal HSIL reduces the progression of HSIL to SCC in people living with HIV (PLHIV) [6]. This observation supports the call for screening programs, likely using HPV DNA testing and/or cytology testing of anal swabs to identify those individuals at greatest risk of HSIL, requiring further investigation and subsequent treatment.

Barriers to obtaining clinician-collected anal swabs (CCA) include the need for an intimate examination, and stigma related to HIV status, sexual orientation and sexual practices, and anal cancer [1, 9, 10]. Alternative approaches to collecting anal swab samples are needed to ensure a high participation and retention rate in future anal cancer screening programs [11]. The use of self-collected anal swabs (SCA) is now well established and validated in the diagnosis of anal sexually transmitted infections (STIs) in GBM using nucleic acid amplification technologies [12], and has been shown to generate technically adequate anal swab specimens in a manner that is acceptable to GBM [11, 13, 14]. The use of SCA eliminates the need for an anal examination at the screening stage, and potentially could be performed at non-clinical sites, subject to further validation. This may be particularly valuable in accessing vulnerable and stigmatised individuals, and in reducing the time pressure on busy clinicians [11].

Here we report on the findings of the randomised clinical Trial of Individually Collected Anal Testing (TICAT), where we compared the diagnostic capabilities of SCA and CCA, in the detection of anal HR-HPV DNA and cytological abnormalities, and the acceptability of both swab collection methods.

## Materials and methods

### Study setting and participants

This was a single site, paired, random sequence clinical trial in which all participants underwent both swab collection techniques at the same visit. Eligible participants were patients attending a routine appointment for monitoring and management of their anal dysplasia at the Dysplasia and Anal Cancer Services (DACS) at St Vincent's Hospital, Darlinghurst, Sydney, and undergoing a high resolution anoscopy (HRA). Patients who did not understand English or were unwilling or unable to provide informed written consent were ineligible for the study. Participants were recruited between 1 December 2020 and 18 August 2022.

### Randomisation and masking

Following recruitment, each participant was randomised in a 1:1 ratio to either the SCA or CCA first and the other technique second. The randomisation sequence was generated and assigned to each participant number before study recruitment.

The most severe abnormality found on either the CCA or SCA was used to inform the subsequent management of the participants and their next standard of care follow-up appointment.

Participant blinding was not possible in the clinic due to the obvious nature of the collection procedure. Laboratory staff were blinded to the swab collection order and collection method during sample analysis.

## Clinic procedures

**Anal swab collection.** For the SCA, the study nurse/doctor verbally explained the technique and provided the participant with an instruction sheet (**S1 Fig**). The participant was issued with a pre-moistened FLOQSwab® 5U002S (COPAN Diagnostics, Murrieta, CA) swab with a flange, to reduce the risk of inserting the swab too far, within the manufacturer-provided container. They were instructed to insert the swab 3–5cm into their anal canal, then firmly rotate and dab it against the walls of the anal canal for about one minute. Patients collected the sample in a private bathroom in the clinic. Immediately after sampling, the swab was briskly swirled by the study nurse/doctor into a ThinPrep® vial containing 20ml of PreservCyt® fixative medium (Hologic Inc, Marlborough, Mass) and then the swab was discarded. The same procedure was followed for the CCA. All staff were trained in the collection of swabs and had substantial prior experience in performing anal swabs for cytology and HPV DNA testing. Both SCA and CCA were collected prior to the start of the HRA procedure.

**Acceptability questionnaire.** Immediately following the clinic procedures, participants were asked to complete a paper questionnaire regarding the acceptability of, and any preference for the sampling techniques (**S2 Fig**). Questions were asked about discomfort or pain experienced with each swab collection methods, satisfaction with SCA instructions, and preference for swab collection method.

## Sample analysis

Both SCA and CCA ThinPrep® sample vials were sent to SydPath, St Vincent's Pathology, Darlinghurst, Sydney, where a portion of the vial liquid was aliquoted into a second vial. One vial was tested for HPV DNA at SydPath, and the other was sent to Douglass Hanly Moir Pathology for cytology testing.

**HPV DNA testing.** ThinPrep® samples were tested for HPV DNA using the COBAS 4800 system (Roche Molecular Systems), which provides partial genotyping, stratifying results into HPV16/HPV18, and other HR-HPV (genotypes 31, 33, 35, 39, 45, 51, 52, 56, 58, 59, 66 and 68). A ß-globin internal cellular control was used to indicate the validity of the sampling. An invalid sample was either due to insufficient cell mass, or to PCR inhibiting substances in the sample.

**Anal cytology.** A ThinPrep® slide (Hologic Inc, Marlborough, Mass) was produced in accordance with manufacturer's instructions, and screened by an experienced cytologist. Anal cytology results were classified according to the Bethesda System as: Unsatisfactory for evaluation, Negative for intraepithelial lesion and malignancy (NILM); Atypical squamous cells of uncertain significance (ASC-US); Low-grade Squamous Intraepithelial Lesions (LSIL); Atypical squamous cells, cannot exclude high-grade squamous intraepithelial lesion (ASC-H), High-grade Squamous Intraepithelial Lesion (HSIL)–Anal Intraepithelial Neoplasia (AIN)2; HSIL-AIN3; and squamous cell carcinoma (SCC) [15, 16]. Slides with fewer than 2000 nucleated squamous cells and no abnormal cells were classified as unsatisfactory for evaluation. Slides with fewer than 2000 nucleated squamous cells but with a cellular abnormality were

reported according to the level of abnormality (as above), but were noted to be 'of low cellularity' for the purposes of analysis.

## Data analysis

The proportions of valid samples for HPV DNA and cytological assessment and their 95% confidence intervals were calculated using Stata (version 18, StataCorp LP, College Station, Texas). HPV DNA test results were categorised as "Any HPV", HPV16 only, HPV18 only, or "Other [non-16/18] HR-HPV") for analysis. Cytology results were categorised as "any abnormality", which was any satisfactory sample result other than NILM, or as "HSIL/ASC-H" (versus LSIL/ASC-US/NILM) for analysis.

The Bethesda system notes that if a sample has low cellularity (and would usually be classified as 'unsatisfactory'), but an abnormal cell(s) is seen in the sample, it is reclassified as 'adequate', and the abnormal result recorded. Since we wanted to compare true cellular adequacy (regardless of cytology results), for calculations relating to the cellular adequacy of the cytology slide, slides deemed 'unsatisfactory' and those 'of low cellularity' (but an abnormality was seen) were aggregated and denoted 'unsatisfactory' for the purposes of the analysis.

To quantify the diagnostic agreement between SCA and CCA, we used CCA as the "gold standard" to calculate sensitivity, specificity, concordance, and the Kappa statistic.

For comparisons between SCA and CCA collected from the same patient, the McNemar test for paired samples was used to test for statistical significance. For comparison between orders of SCA and CCA, a $\chi^2$ test was used to assess statistical significance.

A fixed effects logistic regression model was used to estimate the effects of swab collection order and technique on the outcome variables of sample validity/adequacy (for HPV DNA and cytology testing), and a positive test result (for presence of any cytological abnormality or HPV DNA).

## Ethical considerations

This study was approved by the Ethics Committee at St Vincent's Hospital, Darlinghurst, Sydney (2020/ETH00767; RGO Number: 2020/STE02308)).

## Results

### Study population

A total of 100 participants were recruited into the study. Eight people who were invited declined to take part (a refusal rate of 7.4%). The majority (94.0%) of participants were male, the median age was 57 years (IQR 50.5–63), 87.0% were GBM, and 58.0% were HIV positive. Most (85.0%) participants had previously undergone HRA (**Table 1**).

Fifty-four participants were randomised to SCA first, and forty-six participants to CCA first. There were no differences in participant characteristics between the randomisation orders, however, more participants had never undergone an HRA previously in the SCA-first group (12 individuals (22.2%, 95% CI 12.9–35.5)) compared with the CCA-first group (3 individuals (6.5%, 95% CI 2.1–18.9, p = 0.028)).

### Anal cytology

There was no significant difference in cellular adequacy for cytology testing for SCA versus CCA (77.0% (95% CI 67.6–84.3) and 84.0% (95% CI 75.3–90.0) respectively, p = 0.162) (**Table 2**).

**Table 1. Demographics of study population.**

| Characteristic | Total<br>n (%, 95% CI)<br>(N = 100) | Self-collected first<br>n (%, 95% CI)<br>(N = 54) | Clinician-collected first<br>n (%, 95% CI)<br>(N = 46) | P-values* |
|---|---|---|---|---|
| **Age (years)** | | | | |
| Median | 57 years (IQR 50.5–63) | 54 years (IQR 49–62) | 58.5 years (IQR 53–64) | 0.087 |
| **Gender** | | | | |
| Man | 94 (**94.0**, 87.2–97.3) | 52 (**96.3**, 86.0–99.1) | 42 (**91.3**, 78.5–96.8) | 0.295 |
| Woman | 6 (**6.0**, 2.7–12.8) | 2 (**3.7**, 0.9–14.0) | 4 (**8.7**, 0.3–21.5) | |
| **Sexuality** | | | | |
| MSM | 87 (**87.0**, 78.8–92.4) | 49 (**90.7**, 79.3–96.2) | 38 (**82.6**, 68.5–91.2) | 0.228 |
| **HIV status** | | | | |
| Negative | 41 (**41.0**, 31.7–51.0) | 21 (**38.9**, 26.7–52.7) | 20 (**43.5**, 29.7–58.3) | 0.642 |
| Positive | 58 (**58.0**, 48.0–67.4) | 32 (**59.3**, 45.5–71.7) | 26 (**56.5**, 41.7–70.3) | 0.782 |
| Unknown | 1 (**1.0**, 0.1–6.9) | 1 (**1.9**, 0.2–12.5) | 0 (-) | 0.354 |
| **Previous HRA** | | | | |
| Yes | 85 (**85.0**, 76.5–90.8) | 42 (**77.8**, 64.5–87.1) | 43 (**93.5**, 81.2–97.9) | **0.028** |
| No | 15 (**15.0**, 9.2–23.5) | 12 (**22.2**, 12.9–35.5) | 3 (**6.5**, 2.1–18.8) | |

*Difference between those assigned to self-collection first versus clinician-collected first. Using $\chi^2$ test for categorical variables and t-test for continuous variables.

Among swabs with cellularly adequate samples, there were no statistically significant differences between SCA and CCA in the proportion of samples with any cytological abnormality, either overall (79.4% versus 82.4%, p = 0.480) (**Table 2**), even after adjustment for swab collection order (**S1 Table**). Similarly, there was no significant difference between SCA and CCA in the proportion of samples reported as cytologic HSIL or ASC-H, versus LSIL or ASC-US or NILM, either overall (38.2% versus 29.4%, p = 0.134) (**Table 2**), or when considering swab order (**S1 Table**).

**Table 2. Comparison between self-collected and clinician-collected anal swabs in cytological assessment.**

| Cytology results | Total (N = 100)<br>n (%, 95% CI) | | |
|---|---|---|---|
| | **SC** | **CC** | *P-value* |
| **Sample adequacy** | | | |
| Satisfactory | 77 (**77.0**, 67.6–84.3) | 84 (**84.0**, 75.3–90.0) | 0.162 |
| **Transformation zone** | | | |
| Detected | 58 (**58.0**, 48.0–67.4) | 54 (**54.0**, 44.1–63.6) | 0.480 |
| **Identified abnormalities*** | | | |
| HSIL | 14 (**20.6**, 12.5–32.1) | 8 (**11.8**, 5.9–22.0) | |
| ASC-H | 12 (**17.6**, 10.2–28.8) | 12 (**17.7**, 10.2–28.8) | |
| HSIL/ASC-H | 26 (**38.2**, 27.3–50.5) | 20 (**29.4**, 19.7–41.5) | 0.134 |
| ASC-US | 11 (**16.2**, 9.1–27.1) | 10 (**14.7**, 8.0–25.5) | |
| LSIL | 17 (**25.00**, 16.0–36.8) | 26 (**38.2**, 27.3–50.5) | |
| LSIL/ASC-US | 28 (**41.2**, 30.0–53.4) | 36 (**52.9**, 40.9–64.6) | 0.088 |
| Any abnormality | 54 (**79.4**, 67.9–87.5) | 56 (**82.4**, 71.2–89.8) | 0.480 |

*Excludes samples that were classed as "unsatisfactory" for sample adequacy by either SCA or CCA or both (denominator = 68 (40 SCA first; 28 for CCA first). HSIL: High-grade squamous intraepithelial lesion; ASC-H: Atypical squamous cells, cannot exclude high-grade squamous intraepithelial lesion; ASC-US: Atypical squamous cells of uncertain significance; LSIL: Low-grade squamous intraepithelial lesion. P values are from McNemar paired case-control study test.

**Table 3. Odds ratios for the independent effects of anal swab collection method (SCA or CCA) and swab collection order for cytological and HPV DNA testing.**

| | Univariable | | | | Multivariable* | | | |
|---|---|---|---|---|---|---|---|---|
| | Collection method (SCA vs CCA) | | Collection order (First vs Second) | | Collection method (SCA vs CCA) | | Collection order (First vs Second) | |
| | OR (95% CI) | P-value | OR (95% CI) | P-value | OR (95% CI) | P-value | OR (95% CI) | P-value |
| *A. Cytology* | | | | | | | | |
| *Cellular adequacy* | 0.64 (0.34–1.20) | 0.163 | 1.21 (0.65–2.27) | 0.550 | 0.63 (0.33–1.20) | 0.152 | 1.26 (0.67–2.38) | 0.478 |
| *Any cytological abnormality* | 1.04 (0.61–1.76) | 0.884 | 0.70 (0.41–1.20) | 0.198 | 1.08 (0.64–1.84) | 0.771 | 0.70 (0.41–1.19) | 0.188 |
| *B. HPV DNA testing* | | | | | | | | |
| *Swab valid* | 1.21 (0.63–2.34) | 0.566 | 1.81 (0.93–3.54) | 0.085 | 1.16 (0.61–2.19) | 0.649 | 1.79 (0.92–3.45) | 0.084 |
| *Positive HPV16 test* | 1.24 (0.99–1.55) | 0.056 | 1.04 (0.83–1.31) | 0.707 | 1.24 (1.00–1.54) | 0.052 | 1.03 (0.83–1.27) | 0.812 |
| *Positive HR-HPV test* | 0.78 (0.59–1.03) | 0.081 | 1.09 (0.82–1.44) | 0.565 | 0.77 (0.59–1.02) | 0.073 | 1.11 (0.84–1.46) | 0.469 |
| *Positive HPV test (any)* | 0.85 (0.63–1.14) | 0.279 | 0.91 (0.67–1.23) | 0.533 | 0.85 (0.64–1.15) | 0.297 | 0.93 (0.69–1.24) | 0.609 |

* Collection method adjusting for collection order; Collection order adjusting for collection method.

**Table 3A** shows the effect of swab collection method and swab collection order on cellular adequacy. There was a slightly lower odds of obtaining an adequately cellular sample for cytology testing by SCA versus CCA after adjusting for collection order, but this was not statistically significant (OR = 0.63 (95% CI 0.33–1.20), p = 0.152). There was no significant difference in the odds of detecting of any cytological abnormality by SCA or CCA after adjusting for collection order (OR = 1.08 (95% CI 0.64–1.84), p = 0.771).

## Anal HPV

There was no significant difference in the proportion of samples valid for HR-HPV DNA testing from SCA and CCA, either overall (94.0% (95% CI 87.2–97.3) and 95.0% (95% CI 88.4–97.9) respectively, p = 0.564) (**Table 4**), or when considering swab order (**S2 Table**).

There was no significant difference in the proportion of samples with any HR-HPV detected between the two swab collection methods, either overall (67.7% versus 71.0%, p = 0.317) (**Table 4**) or when considering swab order (**S2 Table**). Similar findings were also observed for samples with HPV16, or HPV18, or other HR-HPV genotypes.

**Table 4. Comparison between self-collected and clinician-collected swabs in HPV DNA testing.**

| HPV result | Total (N = 100) n (%, 95% CI) | | |
|---|---|---|---|
| | SCA | CCA | *P-value* |
| **Sample validity*** | | | |
| Valid | 94 (**94.0**, 87.2–97.3) | 95 (**95.0**, 88.4–97.9) | 0.564 |
| **HPV present**** | | | |
| Any HR-HPV | 63 (**67.7**, 57.5–76.5) | 66 (**71.0**, 60.8–79.4) | 0.317 |
| HPV16 | 32 (**34.4**, 25.4–44.7) | 28 (**30.1**, 21.6–40.3) | 0.103 |
| HPV18 | 9 (**9.7**, 5.1–17.7) | 8 (**8.6**, 4.3–16.4) | 0.564 |
| Other HR-HPV | 48 (**51.6**, 41.4–61.7) | 54 (**58.1**, 47.7–67.8) | 0.083 |

*Valid test for *all* HPV testing

**Only includes participants where both SCA and CCA tests were valid for a particular HPV subtype (denominator = 93 (52 SCA first; 41 for CCA first); †Non HPV16/18 HR genotype

Prevalence of any HR-HPV was high in this study population, with around 70% of both SCA and CCA testing positive. Prevalence of HPV-16 was 34.4% (95% CI 25.4–44.7) from the SCA, and 30.1% (95% CI 21.6–40.3) from the CCA (p = 0.103). Prevalence of any other HR-HPV DNA was 51.6% (95% CI 41.4–61.7) from the SCA, and 58.1% (95% CI 47.7–67.8) from the CCA (p = 0.083).

Table 3B shows the effect of swab collection method and swab collection order on HPV DNA test validity. There was no significant difference in the odds of obtaining a valid sample for HPV DNA testing by SCA or CCA after also adjusting for collection order (OR = 1.16 (95% CI 0.61–2.19), p = 0.649). There was no significant difference in the odds of obtaining a positive test for any HPV between SCA and CCA after also adjusting for collection order (OR = 0.85 (95% CI 0.64–1.15), p = 0.297).

## Diagnostic agreement between SCA and CCA swab results for HPV DNA and cytology

Table 5 presents a diagnostic comparison between paired SCA and CCA for cytology and HPV DNA testing results, where CCA are used as the gold standard.

Concordance between SCA and CCA results was moderately good for cytology, both in detecting HSIL/ASC-H (concordance = 76.5%; Kappa = 0.479), and in detecting any abnormality (concordance = 88.2%; Kappa = 0.620). However, it was even higher for HPV DNA testing with concordance values between 87–97% for detecting any HPV DNA, HPV16, HPV18, or any other HR-HPV, and Kappa values showing a substantial to almost-perfect agreement between SCA and CCA.

Sensitivity was very high for detecting any cytological abnormality (91.1%), but specificity was lower (75.0%); however, sensitivity was lower for specifically detecting HSIL (75.0%). Sensitivity and specificity were high for HPV16, HPV18, and other HR-HPV, and overall HR-HPV testing (sensitivity 90.9%; specificity 88.9%).

**Table 5. Diagnostic agreement between SCA and CCA swab sample results.**

| | Total (N = 100) n (%, 95% CI) | | | Concordance % (n/N) | Sensitivity % (95% CI) | Specificity % | Kappa** |
|---|---|---|---|---|---|---|---|
| | SC | CC | Both | | | | |
| **Sample adequate/valid** | | | | | | | |
| Cytology | 77 (**77.0**, 67.6–84.3) | 84 (**84.0**, 75.3–90.0) | 68 (**68.0**, 58.1–76.5) | 75.0 (75/100) | | | |
| HPV | 94 (**94.0**, 87.2–97.3) | 95 (**95.0**, 88.4–97.9) | 93 (**93.0**, 85.9–96.7) | 97.0 (97/100) | | | |
| **Cytology abnormalities*** | | | | | | | |
| HSIL/ASC-H | 26 (**38.2**, 27.3–50.5) | 20 (**29.4**, 19.7–41.5) | 15 (**22.1**, 13.6–33.7) | 76.5 (52/68) | **75.0** (64.7–85.3) | **77.1** (67.1–87.1) | 0.479 |
| Any abnormality | 54 (**79.4**, 67.9–87.5) | 56 (**82.4**, 71.2–89.8) | 51 (**75.0**, 63.2–84.0) | 88.2 (60/68) | **91.1** (84.3–97.9) | **75.0** (64.7–85.3) | 0.620 |
| **HPV testing*** | | | | | | | |
| Any HR-HPV | 63 (**67.7**, 57.5–76.5) | 66 (**71.0**, 60.8–79.4) | 60 (**64.5**, 54.2–73.7) | 90.3 (84/93) | **90.9** (85.1–96.8) | **88.9** (82.5–95.3) | 0.773 |
| HPV16 | 32 (**34.4**, 25.4–44.7) | 28 (**30.1**, 21.6–40.3) | 27 (**29.0**, 20.6–39.2) | 93.6 (87/93) | **96.4** (92.7–100.2) | **92.3** (86.9–97.7) | 0.853 |
| HPV18 | 9 (**9.7**, 5.1–17.7) | 8 (**8.6**, 4.3–16.4) | 7 (**7.5**, 3.6–15.1) | 96.8 (90/93) | **87.5** (80.8–94.2) | **97.7** (94.6–100.7) | 0.806 |
| Other HR-HPV | 48 (**51.6**, 41.4–61.7) | 54 (**58.1**, 47.7–67.8) | 45 (**48.4**, 38.3–58.6) | 87.1 (81/93) | **83.3** (75.8–90.9) | **92.3** (86.9–97.7) | 0.741 |

*Diagnostic comparison calculations are made just for paired samples, i.e. where both SCA and CCA were valid either for cytology, or HPV. For HPV 16, 18, and Other HPV, if a sample had an inadequate result for "Any HR-HPV", it was excluded from the calculation

** ≤ 0: no agreement, 0.01–0.20: none to slight, 0.21–0.40: fair, 0.41–0.60: moderate, 0.61–0.80 substantial, 0.81–1.00: almost perfect agreement. HSIL: High-grade squamous intraepithelial lesion; ASC-H: Atypical squamous cells, cannot exclude high-grade squamous intraepithelial lesion.

**Table 6. Acceptability and preference of SCA vs CCA swab.**

| | Collection method, n (%, 95% CI). N = 99* | | |
|---|---|---|---|
| | **SCA** | **CCA** | *P-value* |
| **Did you find the anal swab uncomfortable?**\*\* | | | |
| Not uncomfortable | 90 (**90.9**, 83.3–95.2) | 89 (**89.9**, 82.1–94.5) | 0.796 |
| Uncomfortable | 9 (**9.1**, 4.8–16.7) | 9 (**9.1**, 4.8–16.7) | 1.000 |
| No response | 0 (-) | 1 (**1.0**, 0.1–7.0) | - |
| **Did inserting the anal swab hurt?**\*\* | | | |
| Did not hurt | 92 (**92.9**, 85.8–96.6) | 93 (**93.9**, 87.0–97.3) | 0.739 |
| Hurt | 7 (**7.1**, 3.4–14.2) | 6 (**6.1**, 2.7–13.0) | |
| **Feeling relaxed doing SCA**\*\* | | | |
| Not relaxed | 19 (**19.2**, 12.5–28.3) | Question not asked | - |
| Relaxed | 80 (**80.8**, 71.7–87.5) | | |
| **Satisfaction with SCA instructions**\*\* | | | |
| Satisfied | 97 (**98.0**, 92.2–99.5) | | |
| Dissatisfied | 1 (**1.0**, 0.1–7.0) | Question not asked | - |
| No response | 1 (**1.0**, 0.1–7.0) | | |
| **Overall preference**\*\* | | | |
| SCA | 15 (**15.2**, 9.3–23.7) | | |
| CCA | 35 (**35.4**, 26.5–45.4) | | |
| No preference | 48 (**48.5**, 38.7–58.4) | | |
| No response | 1 (**1.0**, 0.1–7.0) | | |

*One participant did not complete a questionnaire

\*\*"Not uncomfortable" includes responses "Not at all" and "A little"; "Uncomfortable includes responses "A fair bit", and "Quite a lot"; "Did not hurt" includes responses "Not at all" and "A little"; "Hurt" includes responses "A fair bit", "Quite a lot", and "Very much"; "Not relaxed" includes response "Not at all" and "A little"; "Relaxed" includes responses "A fair bit", "Quite a lot", and "Very much"; "Satisfied" includes response "Satisfied" and "Very satisfied"; "Unsatisfied" includes response "Dissatisfied" and "Very dissatisfied"; "SCA" preference includes response "Much preferred SCA" and "Partly preferred SCA"; "CCA" preference includes "Much preferred SCA" and "Partly preferred CCA".

## Acceptability and preference of swab collection method

The great majority of participants did not find the SCA or the CCA uncomfortable (90.9% versus 89.9% respectively, p = 0.796) or painful (92.9% for SCA versus 93.9% for CCA respectively, p = 0.739) (**Table 6**).

Most (80.8%) participants reported feeling relaxed when collecting the SCA, and almost all (98%) of participants were satisfied with the SCA instructions provided.

Overall, 48.5% of participants did not have a preference for either SCA or CCA collection method, with 15.2% preferring SCA, and 35.4% preferring CCA.

## Discussion

Self-collected anal swabs are a proposed alternative to clinician-collected swabs (CCA) to overcome barriers to anal cancer screening, such as the need for an intimate examination, and stigma around HIV status, sexual orientation, and sexual practices. However, the diagnostic capabilities of SCA compared with CCA, in the detection of anal HR-HPV DNA and cytological abnormalities, and the acceptability of both swab collection methods are not well studied, which was the rationale of this study. In a study of patients attending a tertiary hospital for

investigation for anal HPV-associated lesions, self-collected anal swabs provided similar diagnostic capability to those collected by experienced clinicians, for both HR-HPV DNA testing, and cytological testing, and were acceptable to patients. More patients preferred clinician-collected swabs.

A strength of this study is that we randomised the order of swab collection, whereas most other studies comparing SCA with CCA have had a fixed order [17–24], meaning it is difficult to ascertain whether swab order is directly contributing to differences in sample cellularity. The multivariate regression analysis indicated that neither swab method or order significantly impacted outcomes either for HPV or cytological testing.

There was no significant difference in sample validity for HR-HPV DNA testing, or in the prevalence of HR-HPV found by SCA and CCA swabs. Adequacy was close to 95% for both SCA and CCA samples, and HR-HPV prevalence was high at 67.7% for SCA and 71.0% for CCA. This prevalence is consistent with a previously reported HR-HPV prevalence estimate of ~70% in a similar population of GBM in Sydney [25]. Concordance between SCA and CCA swabs for HR-HPV DNA testing was excellent, with a substantial to almost-perfect agreement (Kappa 0.74–0.85). Sensitivity and specificity values for SCA compared with CCA were also high, at 90.9% and 88.9% respectively. These findings are consistent with other studies, where SCA swab and CCA swab results for HPV DNA testing were found to be highly concordant, with good agreement levels between the two collection methods for detecting HR-HPV [11, 19, 20, 22, 24, 26].

Cellular adequacy for cytology testing was slightly lower by SCA than CCA, but this was not statistically significant. Other published studies focussing specifically on SCA versus CCA swabs for cytology testing suggest a similar trend with cellular adequacy. In three studies, adequacy for cytology was high overall for both CCA and SCA, but slightly lower for SCA (99% versus 91% in Cranston *et al*, 100% versus 90% in McNeil *et al*, and 92.7% versus 83.3% in Lampinen *et al*) [18, 27, 28]; Heid-Picard *et al* found a much lower adequacy by SCA versus CCA (63% versus 100%) [20, 27]. Several reasons for these differences are suggested, including the experience of the participants and clinicians with anal sampling, the quality and detail of the instructions given, the HIV status of the participants, and the sample size. The sample collection device used across different studies may also be important in the context of optimising cellular adequacy. A FLOQSwab® flocked swab was used in this study, while the other three studies comparing SCA versus CCA for cytology testing used a Dacron swab. However, evidence suggest that flocked swabs may perform slightly better than Dacron swabs for anal cytology [29, 30].

The great majority of participants (≥90%) did not find either SCA or CCA uncomfortable and did not experience pain on insertion of the SCA or CCA swab. Whilst almost half of participants did not express a preference for either collection method, 35% of participants preferred the CCA, compared with 15% preferring the SCA. Data were not collected on the reason for these responses. It could be that this response reflects the setting in which the study was conducted, where participants may have received a CCA prior to the study, were physically present in a tertiary referral centre and may want the "best test" from a highly experienced clinician. Few other studies have looked at acceptability and preference of SCA for anal cancer screening. One study conducted in women suggested a preference for CCA [21], another study in HIV-positive women suggested no preference for either method [28], and a study in ethnic minority persons living with HIV showed a preference for SCA [23]. In a study examining acceptability of SCA in GBM, "not doing the test right" and "the test might not be accurate" were common concerns around self-administered swabs [31]. In the STI testing context, SC anal or rectal swabs are generally preferred over CC swabs. It may be that performing a SCA for anal cancer screening several times will overcome concerns about not performing the SCA correctly.

The main limitation of this work is the small sample size, like most other studies comparing SCA and CCA, resulting in limited study power. However, despite the small sample size, we were still able to gain valuable insights into the comparison between SCA and CCA sampling methods. In addition, the study population is not fully representative of the population who would be candidates for anal cancer screening. Participant recruitment was via a convenience clinic-based sample, and the vast majority had undergone at least one HRA previously; despite the randomisation of participants there was also a significant difference in the proportion of participants who had previously undergone an HRA between the SCA and CCA group. Participants were more used to having anal swabs taken by a clinician, and so may therefore have a greater understanding of the process of self-sampling. Participants were also mainly older GBM, though whilst this limits generalisability, GBM are most likely to be a first target group for future anal cancer screening programs. The involvement of the clinical trial coordinator in assisting the participant with their SCA (by pre-moistening the swab, and by swirling the swab in the ThinPrep® medium after sample collection) does not mirror a real-world scenario where a participant would likely have to do these steps themselves.

Further work should focus on how to improve the quality of the swab sample taken for cytology, such as better instructions, the type of swab used (including the presence or absence of a flange), more rotations of the swab, and collecting multiple swabs per tube [24, 32]. It is likely that, in at-risk populations where anal screening would be more frequent, the quality of sample collection will improve with practice, and more frequent screening will also reduce the likelihood of missing cellular abnormalities. Further work is needed to establish the most suitable periodicity of screening, and location of self-collection, such as home-based or clinic-based self-testing. Studies of SCA in other at-risk populations with lower prevalence of HPV and anal lesions, such as women living with HIV and solid organ transplant recipients, should also be undertaken.

The recent publication of international clinical guidelines for screening of those at risk of ASCC recommends screening approaches including the collection of anal swabs for detection of HR-HPV DNA, and/or cytological abnormalities as a triage to HRA [33]. Our study suggests that self-collected swabs may be a low-cost, effective, feasible, and acceptable alternative to clinician-collected swabs for anal cancer screening, but larger screening studies are needed.

## Supporting information

**S1 Table. Comparison between self-collected and clinician-collected anal swabs in cytological assessment.**
(DOCX)

**S2 Table. Comparison between self-collected and clinician-collected swabs in HPV DNA testing.**
(DOCX)

**S1 Fig. Instructions for how to perform a self-collected anal swab (SCA).**
(TIF)

**S2 Fig. Acceptability questionnaire.**
(PDF)

## Acknowledgments

We acknowledge Simon Comben, Holly Austine and Dan Seeds for their role in supporting this work at St Vincent's Hospital DACS clinic, where this study was conducted.

## Author Contributions

**Conceptualization:** Richard J. Hillman.

**Data curation:** Fengyi Jin.

**Formal analysis:** Clare E. F. Dyer, Fengyi Jin.

**Funding acquisition:** Richard J. Hillman.

**Investigation:** Jennifer M. Roberts, Annabelle Farnsworth, Leon P. McNally, Richard J. Hillman.

**Methodology:** Fengyi Jin, Jennifer M. Roberts, Leon P. McNally, Richard J. Hillman.

**Project administration:** Richard J. Hillman.

**Resources:** Jennifer M. Roberts, Annabelle Farnsworth, Richard J. Hillman.

**Supervision:** Richard J. Hillman.

**Visualization:** Fengyi Jin.

**Writing – original draft:** Clare E. F. Dyer.

**Writing – review & editing:** Fengyi Jin, Jennifer M. Roberts, I. Mary Poynten, Philip H. Cunningham, Andrew E. Grulich, Richard J. Hillman.

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
