## [Decision Letter · Decision Letter 0]

24 Jun 2024

PONE-D-24-14018Self- Versus Clinician-Collected Swabs in Anal Cancer Screening: A Clinical TrialPLOS ONE

Dear Dr.  Dyer,

Thank you for submitting your manuscript to PLOS ONE. After careful consideration, we feel that it has merit but does not fully meet PLOS ONE’s publication criteria as it currently stands. Therefore, we invite you to submit a revised version of the manuscript that addresses the points raised during the review process.

In addition tom the reviewers comments, please perform editing of the discussion part with require corrections. the following outline is recommended for the discussion psrt:Discussion

1.1 Rationale of the study (why it was done)

1.1.1 Main findings of the study

1.1.2 What makes your study unique

1.1.3 What it adds to what we already know

1.2 Subject of the discussion

Comparison of your results with previour similar stidies. Agreement and disagreement with the studies compared

1.3 Summ up of the study, study strengths and limitations

1.4 Clinical implication

We look forward to receiving your revised manuscript.

Kind regards,

Gulzhanat Aimagambetova

Academic Editor

PLOS ONE

Journal Requirements:

2. Thank you for stating the following financial disclosure: "We would like to thank the St Vincent’s Clinic Foundation for funding this project."

3. Thank you for stating the following in the Acknowledgments Section of your manuscript: "We acknowledge Simon Comben, Holly Austine and Dan Seeds for their role in supporting this work at St Vincent’s Hospital DACS clinic, where this study was conducted."

Please remove any funding-related text from the manuscript and let us know how you would like to update your Funding Statement. Currently, your Funding Statement reads as follows: "We would like to thank the St Vincent’s Clinic Foundation for funding this project."

Reviewers' comments:

Reviewer's Responses to Questions

**Comments to the Author**

1. Is the manuscript technically sound, and do the data support the conclusions?

Reviewer #1: Yes

Reviewer #2: Yes

2. Has the statistical analysis been performed appropriately and rigorously? 

Reviewer #1: Yes

Reviewer #2: I Don't Know

3. Have the authors made all data underlying the findings in their manuscript fully available?

Reviewer #1: Yes

Reviewer #2: Yes

4. Is the manuscript presented in an intelligible fashion and written in standard English?

Reviewer #1: Yes

Reviewer #2: Yes

5. Review Comments to the Author

Reviewer #1: The present article is a randomized clinical trial comparing adequacy and acceptability of self vs clinician-collected swab for HPV DNA an cytology test in anal cancer screening. It is a very interesting topic, with little data provided by a randomized study. Furthermore, few previous studies evaluated citology and HPV DNA at the same time.

A very interesting and original aspect of the study is the order randomization. The paper is very well written.

I consider that the article provides relevant and original data to be considered for publication in PLOS ONE. However, some comments should be addressed

Comments:

1- Sample size: could the authors explain how to calculate the sample size?. The small sample used could result in limited study power as autor acknowledges in the limitations. On the other hand there is a great heterogeneicity of participants, in terms of sex and HIV serostatus. Both variables influence the prevalence of HPV and citological abnormalities and should be analyzed separately. Could the authors explain how hetherogeneicity could be addressed; perhaps women and heterosexual men could be excluded or analyzed separately?

2- Tha authors explain in methods that participants underwent a high resolution anoscopy. Why did the authors not use HRA as the gold standard for citology accuracy (sensitivity and specificity) of the two swabs?. I suggest adding this limitation in case authors are unable to performs the aforementioned analysis.

3- Agreement between swabs is good, except for adequate citology sample, which may be associated with decreased HSIL/ASC-H sensitivity. What do the authors think about this topic? Could the authors add this information in the text and give their opinion.

3- Table 3A: I suggest using the same words as in the text: Swab adequate to cellular adequacy; Non-negative diagnosis to Any cytological abnormality.

Reviewer #2: The authors report the findings of a clinical trial evaluating the concordance in HPV detection and anal cytology on testing self-collected as compared to clinician-collected anal swabs. The results of this small study are interesting and add to previously published reports, however the authors need to clarify and/or address the following points:

1) Table 5 – Results’ concordance should include not only in percentages but also the denominators/numbers used. Authors should also clarify in the Results how they assessed Sensitivity and Specificity as this was not described in the text nor in the legend, particularly as anal histology was not available to define “diseased” and “non-diseased” individuals for clinical sensitivity and specificity assessment.

2) Several tables in the manuscript report overall results of self-collected and clinician-collected samples as well as separately according to the order of collection. As no statistical difference was found in the results based on the order of collection, as shown in Table 3, the authors should simplify Tables 2, 4 and 5 by reporting only the overall test results on SCA and CCA. Additional information on collection order could be included as supplementary tables.

3) “Discussion” page 15 lines 242-249: The authors, in comparing their results of sample adequacy of SCA and CCA with those reported in previously published studies should also include the potential role of different “sample collection devices” used in the previous reports. Different preanalytical and analytical workflows must be considered when comparing the results of different studies.

4) “Data analysis” page 6 lines 129-130: Authors state that for cellular adequacy, unsatisfactory anal cytology samples and those of “low cellularity” were aggregated. It would be of value to report the number (%) of each of these two categories in the text and whether lower HPV positivity rates were observed in “low cellularity” samples.

5) “Study participants” page 4: Reasons for patients’ referral to “routine appointments” at the DACS Clinic should be better clarified in this section.

6) The Discussion should underline that a larger study population would be required to assess diagnostic accuracy of cytology and/or HPV testing in SCA as compared to CCA, to determine relative clinical sensitivity and specificity.

6. PLOS authors have the option to publish the peer review history of their article (what does this mean?). If published, this will include your full peer review and any attached files.

Reviewer #1: No

Reviewer #2: No

---

## [Author Response · Author response to Decision Letter 0]

10 Sep 2024

Please find responses to the reviewers' and editorial comments in the "Response to Reviewers" file in the submission.

---

## [Decision Letter · Decision Letter 1]

14 Oct 2024

Self- versus clinician-collected swabs in anal cancer screening: A clinical trial

PONE-D-24-14018R1

Dear Dr. Clare E. F. Dyer,

We’re pleased to inform you that your manuscript has been judged scientifically suitable for publication and will be formally accepted for publication once it meets all outstanding technical requirements.

Kind regards,

Gulzhanat Aimagambetova

Academic Editor

PLOS ONE

Additional Editor Comments (optional):

Reviewers' comments:

Reviewer's Responses to Questions

**Comments to the Author**

1. If the authors have adequately addressed your comments raised in a previous round of review and you feel that this manuscript is now acceptable for publication, you may indicate that here to bypass the “Comments to the Author” section, enter your conflict of interest statement in the “Confidential to Editor” section, and submit your "Accept" recommendation.

Reviewer #1: All comments have been addressed

Reviewer #2: All comments have been addressed

2. Is the manuscript technically sound, and do the data support the conclusions?

Reviewer #1: Yes

Reviewer #2: Yes

3. Has the statistical analysis been performed appropriately and rigorously? 

Reviewer #1: Yes

Reviewer #2: Yes

4. Have the authors made all data underlying the findings in their manuscript fully available?

Reviewer #1: Yes

Reviewer #2: Yes

5. Is the manuscript presented in an intelligible fashion and written in standard English?

Reviewer #1: Yes

Reviewer #2: Yes

6. Review Comments to the Author

Reviewer #1: (No Response)

Reviewer #2: The authors have reviewed the manuscript according to the reviewers' comments and to some degree responded to the suggested changes.

7. PLOS authors have the option to publish the peer review history of their article (what does this mean?). If published, this will include your full peer review and any attached files.

Reviewer #1: No

Reviewer #2: No

---

## [Editor Report · Acceptance letter]

6 Nov 2024

PONE-D-24-14018R1 

PLOS ONE

Dear Dr. Dyer, 

I'm pleased to inform you that your manuscript has been deemed suitable for publication in PLOS ONE. Congratulations! Your manuscript is now being handed over to our production team.

Kind regards, 

on behalf of

Dr. Gulzhanat Aimagambetova 

Academic Editor

PLOS ONE